# Preclinical Efficacy and Involvement of AKT, mTOR, and ERK Kinases in the Mechanism of Sulforaphane against Endometrial Cancer

**DOI:** 10.3390/cancers12051273

**Published:** 2020-05-18

**Authors:** Rajani Rai, Kathleen Gong Essel, Doris Mangiaracina Benbrook, Justin Garland, Yan Daniel Zhao, Vishal Chandra

**Affiliations:** 1Stephenson Cancer Center, University of Oklahoma Health Sciences Center, Oklahoma City, OK 73104, USA; rrai@ouhsc.edu (R.R.); Doris-Benbrook@ouhsc.edu (D.M.B.); Justin-Garland@ouhsc.edu (J.G.); 2Division of Gynecologic Oncology, Department of Obstetrics and Gynecology, College of Medicine, University of Oklahoma Health Sciences Center, Oklahoma City, OK 73104, USA; Kathleen-Essel@ouhsc.edu; 3Biostatistics & Epidemiology, College of Public Health University of Oklahoma Health Sciences Center, Oklahoma City, OK 73104, USA; daniel-zhao@ouhsc.edu

**Keywords:** endometrial cancer, sulforaphane, PI3K/AKT/mTOR and MEK/ERK pathway, apoptosis, epithelial to mesenchymal transition, G_2_/M cell cycle arrest

## Abstract

Sulforaphane exerts anti-cancer activity against multiple cancer types. Our objective was to evaluate utility of sulforaphane for endometrial cancer therapy. Sulforaphane reduced viability of endometrial cancer cell lines in association with the G_2_/M cell cycle arrest and cell division cycle protein 2 (Cdc2) phosphorylation, and intrinsic apoptosis. Inhibition of anchorage-independent growth, invasion, and migration of the cell lines was associated with sulforaphane-induced alterations in epithelial-to-mesenchymal transition (EMT) markers of increased E-cadherin and decreased N-cadherin and vimentin expression. Proteomic analysis identified alterations in AKT, mTOR, and ERK kinases in the networks of sulforaphane effects in the Ishikawa endometrial cancer cell line. Western blots confirmed sulforaphane inhibition of AKT, mTOR, and induction of ERK with alterations in downstream signaling. AKT and mTOR inhibitors reduced endometrial cancer cell line viability and prevented further reduction by sulforaphane. Accumulation of nuclear phosphorylated ERK was associated with reduced sensitivity to the ERK inhibitor and its interference with sulforaphane activity. Sulforaphane induced apoptosis-associated growth inhibition of Ishikawa xenograft tumors to a greater extent than paclitaxel, with no evidence of toxicity. These results verify sulforaphane’s potential as a non-toxic treatment candidate for endometrial cancer and identify AKT, mTOR, and ERK kinases in the mechanism of action with interference in the mechanism by nuclear phosphorylated ERK.

## 1. Introduction

Over the past few decades, the incidence and mortality of endometrial cancer have increased, and the age of onset has decreased due to the global increase in the prevalence of obesity in younger women [1,2]. Approximately 57% of all endometrial cancer cases are related to obesity or being overweight [3]. Although patients with early-stage endometrial cancer are efficaciously treated with surgery, with or without chemo-radiotherapy, resulting in five-year survival rates of 80–85% [4], these therapeutic options are limited in younger patients due to the desire to preserve fertility and in obese patients due to poor hormonal sensitivity and surgical outcomes and higher cost of surgical care [5]. Twenty percent to 30% of endometrial cancer patients are diagnosed with the late-stage disease at the time of surgery, with an estimated five-year survival rate of 40–70% for stage III patients, and a very poor five-year survival of 0–10% for stage IV patients [6]. The prognosis for recurrent disease is even poorer with an expected overall survival of only 14–15 months. The current standard of care for patients with advanced or recurrent endometrial cancer is systemic treatment with carboplatin and paclitaxel. A large phase III non-inferiority study performed by the Gynecologic Oncology Group (GOG) compared carboplatin and paclitaxel to a three-drug regimen of paclitaxel, doxorubicin, and cisplatin (TAP) and showed that carboplatin and paclitaxel were not inferior to TAP in terms of progression-free (PFS) and overall survival (OS) [7]. Overall, carboplatin and paclitaxel also had a favorable toxicity profile. The GOG conducted multiple phase II trials of single-agent chemotherapies in the second-line treatment setting, that had response rates all less than 15% (except for paclitaxel in taxane naïve patients, which had a response rate of 27%) [8]. Therefore, there is a great need to develop more effective and novel therapeutic alternatives for patients with advanced or recurrent endometrial cancer and for obese and young endometrial cancer patients. Clinical trials of therapies that target specific molecular abnormalities for endometrial cancer, such as mTOR inhibitors and vascular epidermal growth factor (VEGF) inhibitors, have shown some promise in the second-line setting [9,10].

Sulforaphane is a rational candidate for development as a therapeutic agent for endometrial cancer. This naturally occurring dietary isothiocyanate (1-isothiocyanato-4-(methylsulfinyl) butane) present in cruciferous vegetables (broccoli, cauliflower, cabbage, etc.) [11] has demonstrated significant chemopreventive activity. Epidemiological studies have also shown that sulforaphane consumption is associated with reduced risk of various human cancers, including lung, colon, breast, stomach, and prostate [12,13,14,15,16]. The chemopreventive activity of sulforaphane involves inhibition of phase 1 enzymes (thereby preventing the conversion of procarcinogens to carcinogens), activation of phase 2 enzymes (leading carcinogens detoxification and excretion from the body), and suppression of pro-inflammatory responses [17]. Sulforaphane also exerts direct effects on cancer cells by inhibition of growth, induction of cell cycle arrest, and activation of apoptosis in various types of cancer cells [18]. Also, sulforaphane modulates epigenetic changes that regulate various molecular targets involved in cell proliferation, differentiation, apoptosis, or cell cycle, and viability of cancer stem cells [19]. The chemopreventive and therapeutic effects of sulforaphane have been comprehensively investigated in various cancers in vitro and in vivo [17]; however, it has not yet been studied in endometrial cancers. The safety of sulforaphane and sulforaphane-rich broccoli sprout extract (sulforaphane-BSE) has been tested clinically [20,21], and multiple phase I and phase II trials have been completed evaluating the efficacy of sulforaphane amongst patients with prostate cancer and breast cancer. These promising activities suggest that sulforaphane may also have value for use in endometrial cancer treatment regimens. 

An additional benefit of sulforaphane is that it plays a major role in energy metabolism and can mitigate obesity by increasing energy expenditure [22]. In an animal model, sulforaphane or its precursor (glucoraphanin) enhanced reduction of food intake, adiposity, and weight gain caused by intraperitoneal injection of leptin [23]. Therefore, sulforaphane is a rational candidate for development as a therapeutic agent for endometrial cancer [24,25,26]. However, to the best of our knowledge, none of the studies has reported the therapeutic efficacy or mechanism of sulforaphane in endometrial cancer, which is closely linked with obesity.

The objective of this study was to evaluate the potential utility of sulforaphane in the treatment of endometrial cancer. Using multiple endometrial cancer cell lines, we evaluated the anticancerous activity of sulforaphane and its mechanism of action. Proteomic analyses were performed to identify signaling mechanisms of the sulforaphane activities. Further, an endometrial cancer cell line xenograft study was performed to validate in vitro findings.

## 2. Results

### 2.1. Sulforaphane Inhibits Endometrial Cancer Cell Growth at G2/M Phase

To examine the potential utility of incorporating sulforaphane (Figure 1A) treatment in the care of endometrial cancer patients, we first measured its cytotoxicity against cell lines established from human endometrial cancers possessing a range of cellular differentiation statuses. Use of an MTT (3-(4, 5-dimethylthiazolyl-2)-2, 5-diphenyltetrazolium bromide) assay, which measures mitochondrial metabolism, demonstrated that growth of all the eight cell lines tested were inhibited by sulforaphane in dose- and time-dependent manners (Appendix A). The potency (IC_50_) and efficacy (maximal growth inhibition) values for the 72 h treatment were used to compare the sulforaphane sensitivities of the cell lines Table 1. The Ishikawa cell line was the most sensitive, with a sulforaphane potency of 4.18 µM and efficacy of 90% at 72 h of treatment. Although the ANC3A cell line had a lower potency of 75 nM, the efficacy was the lowest at only 57% upon 72 h of treatment.

For more detailed investigation, we chose the ANC3A, Hec1B, and Ishikawa cell lines to represent a range of sulforaphane sensitivities and tumor grades. All the three cell lines are type-1 endometrial cancer cell lines thus representing the most common form of endometrial cancer. Their hormone receptor and differentiation statuses are representative of the wide range of endometrial cancers. The Ishikawa cell line is well differentiated, expresses the progesterone receptor (PR) and both forms of the estrogen receptor (ER), and is hormone-sensitive. HEC-1B is ER- and PR-negative and moderately differentiated, while AN3CA is ER- and PR-negative and undifferentiated. Because Ishikawa was the most sensitive, we used slightly lower concentrations with a maximum 10 µM of sulforaphane to treat Ishikawa cells in comparison to the doses used for the other two cell lines.

A BrdU incorporation assay demonstrated that sulforaphane-mediated inhibition of cell proliferation involves reduction of cellular DNA replication (Figure 1B,C). ATP assays showed that sulforaphane inhibition of proliferation is associated with reduction in ATP levels (Figure 1D–F). ATP represents the most important form of cellular energy used for various biological processes, and loss of cellular ATP represents significant toxicity. Flow cytometry results demonstrated that sulforaphane treatment led to significant G_2_/M cell cycle arrest (Figure 1G,H). Western blot analysis of key molecules involved in cell cycle regulation showed sulforaphane effects that are consistent with inhibition of proliferation (Figure 1I). Sulforaphane upregulated p21 and p27, which are well known cyclin-dependent kinase (CDK) inhibitors that promote G_2_/M cell cycle arrest via inhibiting the activities of various cyclin-CDK complexes required for progression into M phase [27,28]. p21 is upregulated in early G_2_/M phase and induces arrest in the G_2_ phase [29]. Cdc2/CDK1, another key player in cell cycle regulation, which controls G_2_-M transition in concert with cyclin B, is inhibited by phosphorylation on two different sites, Tyr15 and Thr14 via Wee1, a nuclear kinase, and Myt1 [30,31]. Sulforaphane also increased phosphorylated Cdc2, phosphorylated Wee1, and myelin transcription factor 1 (Myt-1). Cdc2 is phosphorylated in the G_2_ phase, leading to cell cycle arrest, and prevents entry into mitosis but is de-phosphorylated in the M phase leading progression to mitosis [32]. Cyclin B1 is upregulated in late S phase with gradual increases and a peak in the G_2_/M phase, where its complex with Cdc2 drives the transition from G_2_ to M phase [33,34]. Although sulforaphane upregulated cyclin B1 expression, the phosphorylation and inhibition of Cdc2 by sulforaphane appear to prevent the function of cyclin B1/Cdc2 complex in driving progression of mitosis. Further, cyclin A expression was down-regulated in Hec1B and AN3CA cells and upregulated in Ishikawa cells.

### 2.2. Sulforaphane Induces Endometrial Cancer Cell Death

We next evaluated if apoptosis contributes to the sulforaphane mechanism of growth inhibition. At the cellular level, the terminal deoxynucleotidyl transferase dUTP nick end labeling (TUNEL) assay revealed that cells with fragmented DNA were abundant in the sulforaphane treated cultures as compared to control untreated cultures (Figure 2A), while flow cytometry analysis of Annexin V/PI stained cells demonstrated sulforaphane induction of apoptosis (Figure 2B). The JC-1 (tetraethylbenzimidazolylcarbocyanine iodide) assay demonstrated significantly decreased mitochondrial membrane potential (MMP) in sulforaphane treated cells as compared to control, which is an early sign of mitochondrial-mediated apoptosis (Figure 2C). At the molecular level, an ELISA assay revealed significant dose-dependent increases in caspase-3 activation (Figure 2D–F), and western blot analysis confirmed the caspase 3 cleavage (Figure 2G). The western blot analysis also demonstrated that sulforaphane caused caspase 9 and poly-ADP ribose polymerase (PARP-1) cleavages, which are apoptotic markers. Further, anti-apoptotic proteins such as Bcl2 and Bcl-xl were downregulated, while pro-apoptotic proteins such as BAX, as well as the BAX/Bcl2 ratio were upregulated by sulforaphane (Figure 2G, Appendix A). Moreover, sulforaphane treatment caused reduction of cyclooxygenase IV (COX IV), which further confirms the compromised mitochondrial membrane potential and decreased ATP level [35]. COX IV downregulation has been shown as an underlying mechanism to sensitize cells to mitochondria-mediated apoptosis [36]. Taken together, these results demonstrated that sulforaphane induces intrinsic apoptosis in endometrial cancer cell lines.

### 2.3. Sulforaphane Inhibition of the Cancerous Phenotype

Since sulforaphane regulation of cell viability was established, we further explored its potential anti-cancer effects using cell culture assays that model tumor establishment and metastases. A colony formation assay demonstrated that sulforaphane caused significant decreases in the number of colonies (Figure 3A,B) indicating that sulforaphane inhibits anchorage-independent growth, which is considered a representation of tumor-forming capability. Matrigel invasion and wound healing scratch assays revealed that sulforaphane significantly reduced cell invasion (Figure 3C,D) and migration (Figure 3E) in endometrial cancer cell lines. Epithelial to mesenchymal transition (EMT), characterized by the loss of epithelial characteristic (E-cadherin expression), and acquisition of a mesenchymal phenotype (N-cadherin and vimentin expression) is a key step contributing cancer progression by directly inducing tumor invasion [37]. Since sulforaphane demonstrated reduction of cell invasion and metastases, we evaluated the expression of EMT-related markers in endometrial cells treated with sulforaphane. Western blot analysis demonstrated that sulforaphane significantly increased E-cadherin and/or downregulated expression of N-cadherin and vimentin, in a cell line dependent manner (Figure 3F,G). Overall, these molecular events are consistent with sulforaphane inhibition of invasion and migration/EMT.

### 2.4. Involvement of Kinase Pathways in Sulforaphane’s Mechanism of Action

To explore sulforaphane regulated pathways, protein lysates from Ishikawa cultures treated with 5 µM sulforaphane or control solvent in triplicate were evaluated by mass spec and Ingenuity analysis. Forty-seven proteins were identified to be significantly up- or down-regulated in expression by sulforaphane with high confidence (<1% false discovery rate) (Appendix A). Ingenuity analysis categorized the proteins into 5 networks involving cell-to-cell signaling and interaction, cell movement, cancer, molecular transport, cell assembly and organization, cell cycle, and cell movement (Appendix A). The two highest-scoring networks integrated with the AKT and ERK kinases (Appendix A, Figure 4A and Figure 5A). Ingenuity analysis identified MYC, beta-estradiol, lipopolysaccharide, and nitrofurantoin, PD98059 (an ERK inhibitor), D-glucose, sirolimus (a mammalian target of rapamycin/mTOR inhibitor) and RICTOR (a component of mTOR2) as upstream regulators of the sulforaphane expression alterations observed (Appendix A). We, therefore, chose to evaluate the roles of AKT, mTOR, and ERK signaling in the mechanism of sulforaphane in endometrial cancer cell lines.

Phosphoinositide 3-kinase (PI3K)/AKT/mTOR signaling is a major cell survival pathway proven to serve an important role in regulating cell proliferation, the cell cycle, and apoptosis. Various studies suggest that the PI3K/AKT/mTOR pathway is frequently activated in human cancers, including endometrial, and that the kinases involved are rational targets for the treatment of tumors [38]. Furthermore, recent studies demonstrated that, during cell transition from normal to malignant, EMT is regulated by the PI3K/AKT/mTOR signaling pathway [39,40]. Therefore, we postulated that the PI3K-AKT pathway might be the key pathway mediating the sulforaphane mechanism of EMT suppression in endometrial cancer cells and investigated sulforaphane effects on AKT signaling. Sulforaphane suppressed phosphorylation of AKT (at Ser473, regulated by PI3K), mTOR (at Ser 2448, regulated by AKT), ribosomal protein S6 kinase (S6K at Thr 389, downstream of mTOR), and eukaryotic translation initiation gactor 4E binding protein 1 (4EBP1at Thr 37/46, downstream of mTOR). These effects of sulforaphane were associated with the downregulation of cyclin D1 and proliferating cell nuclear antigen (PCNA) expression consistent with inhibition of cell proliferation. Altogether, these findings suggest sulforaphane-mediated inhibition of the PI3K/AKT/mTOR pathway is associated with endometrial cell growth suppression (Figure 4A,B).

To assess the involvement of AKT kinase pathways in the mechanism by which sulforaphane reduces the viability of endometrial cancer cell lines, we tested if specific kinase inhibitors could alter the phosphorylation events in conjunction with the cellular consequences. To test this, endometrial cancer cells were first treated with PI3K (200 µM of LY294002) and mTOR inhibitors (200 µM of Torin2) to reduce their respective downstream effector (pAKT ^Ser473^ and p4EBP1^Thr 37/46^, respectively) kinase activity (Figure 4C) followed by treatment with sulforaphane. Pretreatment with these kinase inhibitors significantly reduced viability of all three cell lines, verifying the critical role of these kinases in cell survival (Figure 4D–I). Subsequent sulforaphane treatment had only minor effects or could not further reduce viability (Figure 4D–I). The kinase inhibitor pretreatment abolished the sulforaphane mediated dose-dependent decreases in endometrial cancer cell viability (Figure 4D–I). Taken together, these results demonstrate a critical role of AKT inhibition in the sulforaphane mechanism of action accounting for endometrial cancer cell growth inhibition.

PI3K-AKT-mTOR signaling also crosstalks with ERK/MEK signaling to regulate cell viability [41]. ERK/MEK is another major signaling pathway, which regulates cell proliferation, metastasis, and survival in various cancers. Unlike other kinase phosphorylation events, phosphorylation of ERK via sulforaphane has been shown to contribute to inhibition of cancer cell growth and metastasis in different cancer types [42,43,44]. Therefore, we evaluated the effects of sulforaphane on ERK/MEK signaling. Consistent with other studies, we found that sulforaphane significantly increased phosphorylation of MEK (Ser217/221) (Appendix A) and ERK (Thr202/Tyr204), while the expression of total MEK and ERK were decreased (Figure 5A,B). 

To directly test the role of ERK in the mechanism of sulforaphane, we used a dose (10 µM) of the ERK inhibitor, U0126 (U0), which significantly suppressed phosphorylation of ERK (Thr202/Tyr204) (Figure 5C). In the AN3CA cell line, U0 interfered with sulforaphane dose-responsive reduction in cell viability and ERK phosphorylation (Figure 5D,G,H). U0 was more potent than sulforaphane in the Hec1B cell line, thus preventing further reduction of cell viability and induction of ERK phosphorylation by sulforaphane (Figure 5E,G,H). In the Ishikawa cell line, U0 had minimal effects on cell viability and did not interfere with sulforaphane reduction in cell viability despite its inhibition of sulforaphane induced ERK phosphorylation (Figure 5F–H). U0 did not interfere with sulforaphane induction of PARP1 cleavage in any of the cell lines (Figure 5G). In Ishikawa cells, sulforaphane’s dose-dependent inhibition in cell viability even in the presence of U0 (Figure 5F) suggested that ERK is not the dominant pathway contributing sulforaphane induced cell death in this cell line. To further explore the non-canonical role of phosphorylated ERK (pERK), we studied sub-cellular localization of pERK, since nuclear translocation of pERK has been shown to induce cell survival signaling while cytoplasmic retention contributes to pERK dependent cell death [45,46]. Western blot and densitometric analysis revealed that sulforaphane significantly increased pERK levels in the cytoplasm of all three cell lines, but increased pERK in the nucleus in only the two cell lines (AN3CA and Ishikawa), in which sulforaphane could still cause reduction in viability despite U0 pretreatment. In the Hec1B cell line, the lack of sulforaphane-induced nuclear pERK accumulation (Figure 5I) along with the greater sensitivity of this cell line to U0 in comparison to the other two cell lines, and the inability of U0 to interfere with sulforaphane cytotoxicity (Figure 5D–F), suggest that Hec1B cells are not capable of eliciting a nuclear pERK survival response to prevent cell death. Altogether, these results suggest that ERK phosphorylation contributes to sulforaphane mediated reduction in cell viability, while the accumulation of nuclear pERK interferes with sulforaphane activity in a cell line–specific manner. 

### 2.5. Sulforaphane Inhibits Tumor Growth in Ishikawa Xenograft Study

To validate in vivo inhibition of endometrial cancer growth, sulforaphane effects alone and in combination with the standard of care drug, paclitaxel, were evaluated in an Ishikawa xenograft tumor model. Sulforaphane (50 mg/kg daily), paclitaxel (10 mg/kg once every seven days), or a combination were administered via intraperitoneal injection to tumor-bearing mice (N = 10 per group). When control tumors reached 1000 mm^3^ in diameter, mice were euthanized, and their tumors evaluated. Sulforaphane as a single agent and when combined with paclitaxel induced significant reductions in tumor volume on and after day 24 (Figure 6A, Appendix A). At the doses used, sulforaphane demonstrated greater activity compared to paclitaxel alone, which caused a lower amount of tumor volume reduction that was not significant until day 28. The extent of tumor volume reduction by sulforaphane alone was too high to detect synergy with paclitaxel, however, we can conclude that there was no antagonism. Mouse body weights were consistent throughout the study demonstrating no gross toxicity at the indicated dose regimens (Figure 6B). To access apoptosis as the mechanism of cell death contributing reduction in tumor volume, the TUNEL assay was performed on paraformaldehyde-fixed xenograft tumor tissue and confirmed induction of apoptosis in all treated tumors (Figure 6C). 

## 3. Discussion

In this study, for the first time, we report the efficacy of sulforaphane against endometrial cancer. Sulforaphane inhibited the growth of multiple endometrial cancer cell lines at micromolar concentrations. This growth inhibition involved G_2_/M cell cycle arrest, consistent with reported observations in acute lymphoblastic leukemia, ovarian cancer and colon cancer [15,47,48]. G_2_/M cell cycle arrest is a checkpoint induced by DNA damage that is controlled by a p21/cdc2/cyclin B1 [15,21,28]. In endometrial cancer cell lines, sulforaphane induced phosphorylation of cdc2 on Tyr15, which is consistent with inhibition of the p21/cdc2/cyclin B1 complex driving G2/M progression. Flow cytometric analysis of Annexin V/PI and TUNEL assays and western blot analysis suggest sulforaphane activation of apoptosis in endometrial cancer cells. Sulforaphane has been shown to synergize with cisplatin to suppress cancer cell proliferation and enhance apoptosis in ovarian cancer cells [49]. In this study, sulforaphane inhibited the anchorage-independent growth, migration, invasion, and EMT transition of endometrial cancer cell lines, which are considered as characteristics of neoplastic and metastatic propensity. Overall, our results are consistent with sulforaphane anti-cancerous activity and mechanisms reported for other cancers [42,50].

Aberrant AKT signaling has been frequently reported in various cancers and most often associated with tumor aggressiveness [51]. The Cancer Genome Atlas analysis demonstrated increased AKT activity in more than 90% of endometrial tumors [52]. AKT signaling promotes cell survival by mediating the effects of cellular growth factors, and blocks apoptosis by inactivating pro-apoptotic proteins [53]. Inhibition of AKT phosphorylation has shown promising results in restraining endometrial cell proliferation [54,55]. Based on its role in cancer progression, various inhibitors of PI3K/AKT/mTOR are currently being evaluated in clinical trials for treating patients with metastatic, recurrent, and persistent endometrial cancer. However, the results are inconclusive [56,57]. Our proteomic analysis identified AKT/mTOR signaling alterations in the network of effects induced by sulforaphane in Ishikawa cells. As in other studies [49,58,59], in this paper, we identified sulforaphane as a potent inhibitor of PI3K/AKT/mTOR pathway signaling in endometrial cancer. Sulforaphane downregulated PI3K targeted protein phosphorylation events, such as p-AKT^Ser473^ p-mTOR^Ser 2448^, S6K^Thr 389^, 4EBP^Thr 37/46^, and proliferative markers such as cyclin D1 and PCNA. PI3K/AKT/mTOR inhibition via various molecules or compounds have been shown to induce cancer cell apoptosis through reduction of mitochondrial membrane potential, inactivation of XIAP, activation of caspase-3, upregulation of BAX and downregulation of Bcl-2 [60,61,62]. In our study, sulforaphane treatment demonstrated induction of intrinsic apoptosis in association with upregulation of BAX, and caspase-3 and -9 activity, along with downregulation of anti-apoptotic proteins, such as Bcl-2 and Bcl-xl. This is consistent with the sulforaphane mediated inhibition of PI3K/AKT/mTOR contributing to apoptosis in endometrial cancer cells. Using inhibitors of AKT and mTOR, we demonstrated that these kinases play crucial roles in endometrial cancer cell survival and without them, sulforaphane can no longer reduce endometrial cancer cell line viability.

Sulforaphane has been shown to inhibit cancer cell growth and invasiveness by activating ERK signaling, which in turn regulates apoptosis [45,46,63] and expression of p21, E-cadherin [44], MMP2 and CD44v6 [64]. While cytoplasmic pERK causes cell death, nuclear translocation of pERK induces cell survival signaling [45,46]. Our study demonstrated increased phosphorylation of MEK and ERK by sulforaphane consistent with activation of MEK/ERK signaling in endometrial cancer cells. The ERK inhibitor, U0, interfered with sulforaphane induced phosphorylation of ERK and reduction in cell viability in a cell line dependent manner. U0 pretreatment prevented sulforaphane dose-responsive reduction in cell viability in the Hec1B cell line, which did not exhibit sulforaphane-induced nuclear pERK accumulation. U0 was less effective in countering sulforaphane in the AN3CA cell line and not effective in the Ishikawa cell line, consistent with the accumulation of nuclear pERK upon sulforaphane treatment of these two cell lines. These results indicate that ERK phosphorylation can contribute to the mechanism of sulforaphane reduction in cell viability, while the accumulation of nuclear pERK can interfere with this mechanism.

Significant inhibition of Ishikawa endometrial cancer cell line xenograft growth by sulforaphane in association with induction of apoptosis provided in vivo validation of our cell culture results. Because the utilization of sulforaphane for treatment of endometrial cancer would most likely be tested first in combination with the current standard of care, we evaluated the ability of sulforaphane to enhance paclitaxel reduction of endometrial cancer xenograft dose. The combination of sulforaphane and paclitaxel significantly reduced tumor volume with no gross toxicity as evidenced by no significant differences in body weight gain amongst all treatment groups. Further studies are needed with reduced doses of sulforaphane in combination with paclitaxel in order to observe potential synergy between the two drugs. Also, studies combining sulforaphane with other drugs used to treat endometrial cancer, such as doxorubicin or carboplatin, are warranted. A combination of phytochemicals with chemotherapy and other anticancer agents could be useful since phytochemicals are bioavailable and cause less toxicity compared to chemotherapy. 

## 4. Materials and Methods

### 4.1. Cell Lines, Culture Conditions, and Chemicals

AN3CA, KLE, Hec1A, Hec1B, MFE280, and MFE296 were obtained as a gift from the lab of Jie Wu Ph.D., University of Oklahoma Health Sciences Center, Oklahoma city, OK, 73104, USA. Ishikawa was purchased from Sigma (St Louis, MO, USA). AN3CA, KLE, Hec1B, MFE280, and MFE296 were authenticated by short tandem repeat (STR) profiling at the University of Arizona Genetics Core in 2017 and passaged less than 20 times since authentication. AN3CA, Ishikawa, and Hec-1B were grown in EMEM, MFE280, and MFE296 were grown in DMEM, KLE was grown in DMEM/F12, and Hec1A was grown in McCoy’s 5a. All media was supplemented with 10% FBS and 1% penicillin/streptomycin. Sulforaphane (#S8044; Cas No. 4478-93-7) and paclitaxel (#P0093; Cas No. 33069-62-4) were obtained from LKT laboratories (St. Paul, MN, USA). PI3K inhibitor, LY294002 (#9901), mTOR inhibitor, Torin2 (#14385), and ERK inhibitor U0126 (#9903) were purchased from Cell Signaling Technology (Danvers, MA, USA). In all of the experiments, cells treated with vehicle only were considered as the control and all treatments were given in medium supplemented with 10% FBS. 

### 4.2. MTT Cell Proliferation Assay

Endometrial cancer cells were maintained in their respective media. Human cell lines were removed from culture dishes with 0.05% trypsin-0.02% EDTA, plated at densities of 4000–6000 cells/well in 96 well plates and then incubated at 37 °C for 24 h in a humidified atmosphere with 5% CO_2_ prior to treatment of various concentrations (0–40 µL, 10 concentrations) of sulforaphane. After 24, 48, or 72 h of treatment, 15 µL MTT solution (#G4100, Promega Madison, WI, USA) was added per well and incubated at 37 °C for 1 h, at which point STOP solution was added. After an additional 24 h of incubation, the 96-well plates were measured at the 570 nm wavelength. Graphpad Prism 8 software (San Diego, CA, USA) was used to plot survival (average OD of treated wells/average OD of control wells) versus sulforaphane concentration curves and derive potencies (IC_50_) and efficacies (maximal % growth inhibition) for each cell line and time point. 

### 4.3. BrdU Proliferation Assays

Cell proliferation was measured by using BrdU Cell Proliferation Chemiluminescent Assay (#5492, Cell Signaling, Danvers, MA, USA) that detects BrdU incorporation into cellular DNA during cell proliferation. The experiment was performed according to the manufacturer’s instruction. Briefly, 4000–6000 cells were seeded in 96 well plates and incubated overnight to allow the cells to attach to the bottom of the wells. The cells were then treated in triplicate with various concentrations of sulforaphane for 24 h or 72 h. Finally, BrdU solution was added to each well, and cells were incubated for 4 h. Then the cells were fixed with fixing solution for 30 min at room temperature followed by incubation with 1X detection antibody at room temperature for 1h. After washing, HRP-conjugated secondary antibody was added, and the plate was incubated for 30 min at room temperature. A chemiluminescence substrate solution was then added into each well, and chemiluminescence was measured after 10 min of incubation using a SYNERGY H1 microplate reader (BioTek, Winooski, VT, USA) at a wavelength of 425 nm. The number of proliferating cells is represented by the level of BrdU incorporation, which directly correlates to the absorbance values. 

### 4.4. Mitochondrial Membrane Potential (MMP) Assay

The MMP was analyzed using JC-1 fluorescent dye (5′,6′,-tetrachloro-1,1′,3,3′ tetraethylbenzimidazolyl carbocyanine iodide) and an MMP assay kit (ab113850; Abcam, Cambridge, MA, USA), following the manufacturer’s protocol. Endometrial cancer cells (1.5 × 10^4^ cells/well) were seeded onto a 96-well black plate ((#NC1463153, Perkin Elmer, Waltham, MA, USA) and allowed to adhere overnight followed by sulforaphane treatment as shown. Cells were then washed with 1X dilution buffer provided in the kit and probed with 20 uM JC-1 for 10 min at 37 °C. Subsequently, cells were washed twice with 1X dilution buffer and fluorescence intensity was determined for J-aggregates and J-monomers at excitation and emission wavelengths of 535 nm and 590 nm and 475 nm and 530 nm, respectively. The ratio of J-aggregates and J-monomers was calculated. A decrease in fluorescent intensity ratio is suggestive of Δψ_m._ depolarization.

### 4.5. ATP Assay

Intracellular ATP levels were determined by luminescence using the CellTiter-Glo 2.0 Luminescent Cell Viability Assay (#G9241, Promega, Madison, WI, USA) as per the manufacture’s instruction. Briefly, 1 × 10^4^ cells were seeded into a 96-well black plate and incubated with sulforaphane for 24 h. Then CellTiter-Glo reagent was added to each well, and the contents were mixed for 2 min on an orbital shaker to induce cell lysis, followed by 10 min incubation at room temperature to stabilize the luminescent signal. The luminescence was measured using a SYNERGY H1 microplate reader (BioTek, Winooski, VT, USA) and expressed as fold changes. The luminescent signal is proportional to the amount of ATP in the sample, which indicates the presence of live and metabolically active cells. All values were normalized to protein concentrations to analyze energy change per cell.

### 4.6. Tunnel Assay

To investigate the sulforaphane induction of apoptosis in endometrial cancer cells, the TUNEL assay was performed using DeadEnd Colorimetric TUNEL system (#G7130, Promega, Madison, WI, USA) according to the manufacturer’s protocol. Briefly, the cells were plated in Lab-Tek™ Chamber Slides (#154526, ThermoFisher, Waltham, MA, USA) and incubated overnight. Then the cells were treated either with sulforaphane or vehicle. After 24 h of treatment, cells were fixed in 4% paraformaldehyde in phosphate-buffered saline (PBS) for 25 min at room temperature and treated with 0.2% Triton X-100 for permeabilization followed by PBS washing twice. Then the cells were equilibrated with equilibration buffer at room temperature for 5–10 min and labeled with fluorescein-12-dUTP using terminal deoxynucleotidyl transferase. After stopping the reaction and rinsing with PBS, slides were incubated with 0.3% hydrogen peroxide for 3–5 min followed by PBS Wash (3×) and incubated in Streptavidin HRP (diluted 1:500 in PBS) for 30 min at room temperature. Then, DAB Solution was added to develop color. Slides were mounted with aqueous or permanent mounting medium and observe staining with a light microscope. AN3Ca cell lines treated with 20 µM of cisplatin or vehicle was used as the positive and negative control, respectively. For the tumor tissues, obtained from Ishikawa xenograft study, formalin-fixed paraffin-embedded tissue sections were deparaffinized by incubating slides for 3 min each in 4× Xylol-100%, EtOH-96%, EtOH-80%, EtOH, and 4× H2O. Then, the slides were washed thrice in TBS and incubated with 20 μL proteinase K (10 μg/mL) to completely cover the section for 15 min at room temperature. After washing with TBS, the sections were incubated with 0.2% Triton X-100 and processed as mentioned above.

### 4.7. Annexin-V/PI Apoptosis Flow Cytometry Assay

Apoptosis was detected by Annexin V staining using FITC Annexin V/Dead Cell Apoptosis Kit (# V13242, Thermo Fisher, Waltham, MA, USA) following the manufacturer’s instructions and analyzed by flow cytometry using the FACS Calibur Flow Cytometer (BD Biosciences, San Jose, CA, USA) at core facility. Cells were seeded in a 6-well plate and after overnight incubation, treated with sulforaphane at the specified concentrations. After the 24 h treatments, the cells were harvested and washed with cold PBS. Then, the cells were labeled with Annexin V-PI stain and flow cytometry was performed. In the Annexin V assay, the bottom right quadrant (the percentage of cells positive for only Annexin V staining) and the top right quadrant (cells positive for both Annexin V and PI staining) represent the early and late apoptotic populations, respectively. The cells positive for PI staining only in the top left quadrant are considered to be necrotic. The cells exhibiting double-negative staining in the lower left quadrant are considered to be live cells.

### 4.8. Caspase-3 Activity Assay

Cells were seeded in a six-well plate, and after overnight incubation, cells were treated with sulforaphane or vehicle for 24 h. Afterward, cells were rinsed with PBS and trypsinize to collect cell pellets. Pellets were washed with ice-cold PBS and subsequently cell lysis buffer mPER (#78501, Thermofisher, Waltham, MA, USA) was added to collect cell lysate. Cell lysates were diluted to a concentration of approximately 1mg/ml. Caspase-3 activity was measured using Caspase-3 Activity Assay (Kit #5723, Cell Signaling Technology, Danvers, MA, USA) according to the manufacturer’s instructions. Fluorescence was measured with excitation wavelength at 380 nm and an emission wavelength of 460 nm using a SYNERGY H1 microplate reader (BioTek, Winooski, VT, USA) and expressed in relative fluorescence units [RFU].

### 4.9. Western Blot Analysis

For western blot, cells were seeded in 6-well plates overnight and treated with sulforaphane. After 24 h of treatment with sulforaphane, cells were lysed with mPER (#78501, Thermofisher, Waltham, MA, USA) supplemented with 1% phosphatase inhibitor cocktail and 1% protease inhibitor cocktail. The BCA protein assay was performed using a BCA Assay Kit (#23225, Thermofisher, Waltham, MA, USA) to estimate the total protein concentration. The lysates of equal protein concentrations (30 μg) were then separated by sodium dodecyl sulfate-polyacrylamide gel electrophoresis (SDS-PAGE) and transferred to nitrocellulose membranes using a Trans-Blot^®^ Turbo™ Transfer System. The membranes were incubated in a blocking solution containing 10% nonfat dry milk in Tris-buffered saline containing 0.1% v/v Tween-20 (TBST) for 1h at room temperature. The membranes were incubated with primary antibodies at 4 °C, overnight followed by three times washing with TBST. Then blots were incubated with HRP-conjugated anti-rabbit (dilution, 1:5000) or anti-mouse antibody (dilution, 1:6000) for 45 min at room temperature. After TBST washing, the antibody-protein complexes were visualized with an electrochemical luminescence reagent (Clarity™ Western ECL Substrate) (#1705060S, BioRad, Hercules, CA, USA) according to the manufacturer’s instructions, and subsequent exposure to ChemiDocTM Touch Imaging System (BioRad).

The antibodies Bax (#5023), Bcl-2 (#4223), Bcl-xL (#2764), Cleaved Caspase-9 (#7237), COX IV (#4850), Caspase-3 (#9662), Cleaved Caspase-3 (#9664), Cleaved PARP (#5625), PARP (#9542), Phospho-Wee1 ( #4910), Myt1 (#4282), Cyclin A2 (#4656), Cyclin B1 (#12231), Phospho-cdc2 (#4539), p27 (#3686), p21 (#2947), Phospho-Histone H3 (Ser10) (#3377), Histone H3 (#4499), N-Cadherin (#14215), E-Cadherin (#14472), Vimentin (#5741), Phospho-AKT (#13038), AKT (#4691), Phospho-mTOR (#5536), mTOR (#2983), Phospho-4E-BP1 (#2855), Phospho-p70 S6 ( #9234), PCNA (#13110), Cyclin D1 (#2922), Phospho-p44/42 MAPK (ERK) (#4370), p44/42 MAPK (ERK) (#4695), Phospho-MEK (#9154), MEK (#8727), β-Actin (#4970), Anti-mouse, HRP-linked (#7076), Anti-rabbit, HRP-linked (#7074) were purchased from Cell Signaling Technology (Danvers, MA, USA). Lamin B1 (#66095) antibody was purchased from the vendor Proteintech (Rosemont, IL, USA). Primary antibodies were used in dilutions recommended by the manufacturer. Secondary antibodies were used at a dilution of 1:5000–6000.

### 4.10. Colony Formation Assay

Colony formation assay was carried out in 12-well plates. First, 0.5 mL of 0.6% agar (# 50101, SeaPlaque™ Agarose, Lonza, Basel, Switzerland) in EMEM supplemented with 20% FBS and 1% antibiotics was layered onto the bottom of the plates and allow to solidify for 30 min at room temperature. A total of 2 × 10^3^ endometrial cancer cell lines were then seeded in 0.3% agar containing the desired concentration of sulforaphane or vehicle as the top layer and incubated for 30 min at room temperature to solidify. Approximately 100 µL of culture medium with or without sulforaphane was added to the top layer and the plates were incubated at 37 °C in a humidified atmosphere of 5% CO2. Top Media was replaced with fresh media twice a week. After three weeks, the colonies were scanned for counting the number of colonies formed using Gelcount (Oxford Optronix Ltd, Milton, Abingdon OX14 4SA, United Kingdom) and photographed using an inverted Nikon Eclipse Ni microscope (Minato City, Tokyo, Japan). Each treatment was performed in triplicate.

### 4.11. Invasion Assay

The role of sulforaphane on the invasion capacity of endometrial cancer cells was determined using a matrigel invasion assay and a transwell system (8-µm pore size; # 353097, Corning Inc., Corning, NY, USA). Briefly, 2.5 × 10^4^ cells were seeded onto the matrigel (# 356232 Corning Inc., Corning, NY, USA) coated Transwell chamber (24-well insert) in serum-deprived culture (EMEM supplemented with 5% FBS) and treated with sulforaphane. EMEM supplemented with 10% FBS (as a chemoattractant) was added to the basal chamber. The cells were allowed to migrate for 72 h. Non-migrated cells on the apical side of the chamber were gently scraped off using wetted cotton swabs. The migrated cells at the basal side were fixed using 4% paraformaldehyde and methanol and then stained with 0.5% crystal violet. The assay was performed in triplicate. The images of migrated cells were captured under an inverted microscope using a Nikon Eclipse Ni microscope. The invaded cells were trypsinized and counted using a cell counter (Countess™ II FL Automated Cell Counter, #AMQAF1000, ThermoFisher, Waltham, MA, USA) to quantify the number of migrated cells.

### 4.12. Wound Healing Assay

Endometrial cancer cells were seeded into a six-well assay plate and allowed to grow and form a confluent monolayer at 37 °C in a humidified atmosphere of 5% CO_2_. A scratch or wound was made by using a 10 µL pipette tip, and the plate was washed with PBS to remove cells from a detached area to form a cell-free zone. Cells were treated with sulforaphane, and images of cell movement into wound area were captured at 0 h and 24 h using an inverted Nikon Eclipse Ni microscope.

### 4.13. Tumor Xenograft Model

All studies involving mice were conducted in accordance by a standard animal protocol approved with the Institutional Animal Care and Use Committee (Protocol #18-010-ICH-H), University of Oklahoma Health Sciences Center. Female five-week-old SCID mice were obtained from Taconic Biosciences, Inc and were maintained under a 12-h light/dark cycle under pathogen-free conditions, using sterile food and water within the cage. After acclimation period of 72 h, xenograft formation was generated by the subcutaneous injection of 10 million Ishikawa cells, suspended in 100 µL sterile normal saline. Tumor detection was assessed by palpation and once identified measurement of tumor volume was carried out using calipers three times per week. Tumor volume was calculated utilizing the hemiellipsoid formula Y=πlwh/6 [65]. After tumors reached an average volume of 50 mm^3^, the mice were randomized into four separate groups of 10 mice in each. Sulforaphane was purchased from LKT laboratories and dissolved in sterile normal saline. Paclitaxel was purchased from LKT laboratories and dissolved in cell grade 200 proof ethanol, purchased from Sigma-Aldrich, and then suspended in sterile normal saline. Treatment groups consisted of controls (0.9% normal saline), 50 mg/kg sulforaphane daily, 10 mg/kg paclitaxel weekly, and the combination of 50 mg/kg sulforaphane daily with 10 mg/kg paclitaxel weekly. All treatments were administered via intraperitoneal (IP) administration. Mice were weighed three times a week, and animal health was monitored daily. When the tumor sizes in the control group reached 1000 mm^3^ in diameter after 16 days of treatment, all mice were euthanized by CO_2_ inhalation followed by cervical dislocation to assure death. Tumors were collected at necropsy and a portion of each tumor from each animal was fixed in paraformaldehyde and embedded in paraffin, while another portion was snap-frozen in liquid nitrogen. 

### 4.14. Mass Spectrometry Sample Preparation and LC-MS/MS Measurement

For mass spectrometry analysis, Ishikawa cells were treated with 5 µM of sulforaphane or solvent only for 24 h in triplicate and protein isolates were collected with mPER (#78501, Thermofisher, Waltham, MA, USA). Samples were digested according to the filter aided sample preparation (FASP) protocol as described previously [66]. 

### 4.15. Data Analysis and Statistics

All the experiments were repeated independently three times unless otherwise noted. GraphPad Prism 8 software (GraphPad Software Inc., La Jolla, CA, USA) was used for all statistical analyses. Data were expressed as means ± SD unless otherwise noted. The mean among three or more groups was compared by executing One-way ANOVA and the mean between the two groups was compared by performing Student *t*-test. For all experiments except for the mass spec analysis, statistical significance was set at *p* < 0.05. For the mass spectrometry analysis, proteins that were significantly increased in the triplicate sulforaphane treatments compared to the triplicate control treatments were identified by ANOVA (*p* < 0.05) and the two-stage step-up method of Benjamini, Krieger, and Yekutieli was used to identify high confidence proteins that had <1% false discovery rate. For analysis of the animal model, tumor volume at each time point was compared between the two treatment groups using a linear mixed-effects model for repeated measures.

## 5. Conclusions

In summary, sulforaphane induces promising in vitro and in vivo activity against endometrial cancer cell lines without toxic side effects. These results support the development of sulforaphane addition to the standard of care therapies for advanced and recurrent endometrial cancer. We identified AKT/mTOR inhibition and ERK phosphorylation and nuclear pERK accumulation as key factors that can modulate endometrial cancer cell line sensitivity to sulforaphane. Further studies are warranted to evaluate the mechanisms of these events and determine if modulation nuclear pERK accumulation could enhance sulforaphane anti-cancer activity without enhancing its toxicity.

## Figures and Tables

**Figure 1 cancers-12-01273-f001:**
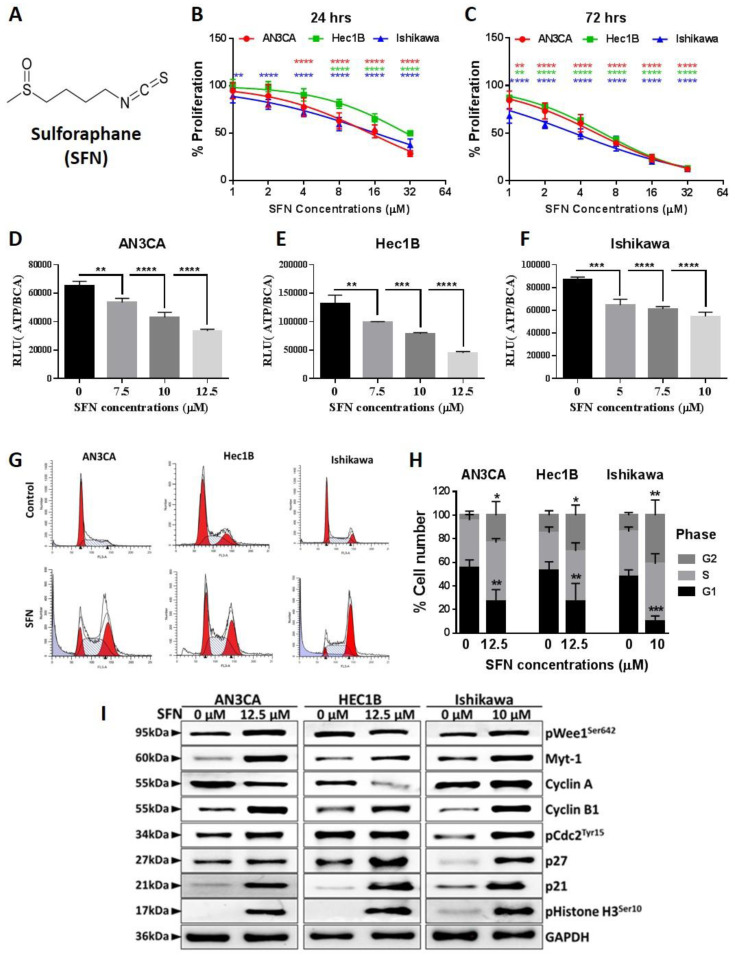
Sulforaphane inhibits endometrial cancer cell viability and induces G_2_/M cell cycle arrest. (**A**) The chemical structure of Sulforaphane. (**B**,**C**) Endometrial cancer cells were treated with sulforaphane at the indicated concentrations for 24 h (**B**) and 72 h (**C**), and cellular proliferation was assessed using the BrdU (bromodeoxyuridine) ELISA (Enzyme-linked immunosorbent assay). Data are the mean ± SD, and two-way ANOVA was used for statistical analysis. (**D**–**F**) Adenosine triphosphate (ATP) levels in sulforaphane treated cells at the indicated concentration for 24 h were measured using Cell Titer Glo Assay. Relative luminescence (RLU) data were normalized to the protein level. Data are expressed as mean ± SD and One-way ANOVA was used for statistical analysis. (**G**,**H**) Endometrial cancer cells were treated with sulforaphane for 24 h, and then cells were stained with propidium iodide and subjected to flow cytometric analysis to determine the cell distribution at each phase of the cell cycle. Representative images of cell cycle distribution with or without sulforaphane were shown (**G**). Percentages of cells at different cell cycle phases are expressed as mean ± SD of three independent experiments and Two-way ANOVA was used for statistical analysis (**H**). (**I**) Cells were treated with sulforaphane for 24 h, and protein isolates were analyzed by western blot with antibodies against cell cycle associated proteins. Glyceraldehyde 3-phosphate dehydrogenase (GAPDH) was used as a reference protein. “*p*” values are indicated * *p* ≤ 0.05, ** *p* ≤ 0.01, ***; *p* ≤ 0.001, **** *p* ≤ 0.0001 when compared with respective control. SFN; sulforaphane. The whole western blot images of Figures please find in Appendix A.

**Figure 2 cancers-12-01273-f002:**
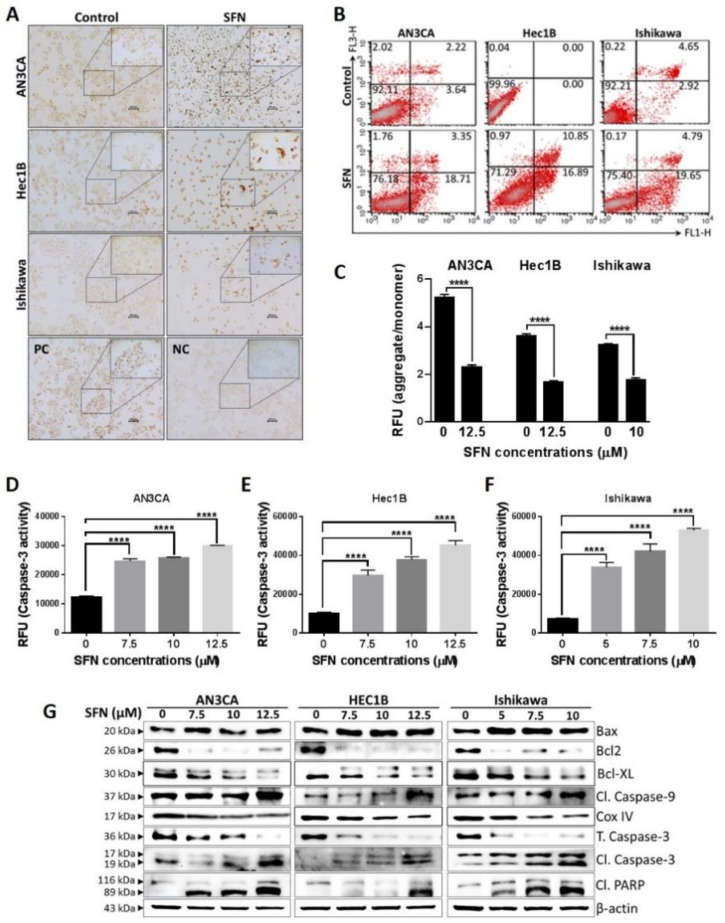
Sulforaphane induces mitochondrial-mediated apoptosis in endometrial cancer cells. (**A**) Endometrial cells were treated with sulforaphane (AN3CA and Hec1B-12.5 µM; Ishikawa-10 µM) for 24 h and TUNEL assay was performed. For positive and negative controls, AN3CA cells were used. Cisplatin (20 µM) was used for the positive control group. Representative images taken at 10× magnification are shown. (**B**) Endometrial cancer cells were treated with sulforaphane (AN3CA and Hec1B-12.5 µM; Ishikawa-10 µM) for 24 h, followed by Annexin V-FITC/PI staining, and the apoptotic rate was analyzed by flow cytometry. Representative images of flow cytometry result with or without sulforaphane were shown. (**C**) JC-1 assay was performed to measure mitochondrial membrane potential in endometrial cancer cells treated with or without sulforaphane for 24 h. A ratio of aggregate and monomer data is expressed as mean ± SD of three independent experiments performed in triplicate. Two-way ANOVA was used for statistical analysis. (**D**–**F**) Endometrial cancer cells were treated with the indicated concentrations of sulforaphane for 24 h, and cleaved caspase-3 level was quantified by a caspase-3 activity assay. Data are expressed as mean ± SD and a one-way ANOVA was used for statistical analysis. (**E**,**G**) Endometrial cancer cells were treated with sulforaphane for 24 h, and the expression of apoptotic protein and anti-apoptotic protein were determined by western blot. “*p*” values are indicated, **** *p* ≤ 0.0001 when compared with respective control. SFN; sulforaphane. The whole western blot images of Figures please find in Appendix A.

**Figure 3 cancers-12-01273-f003:**
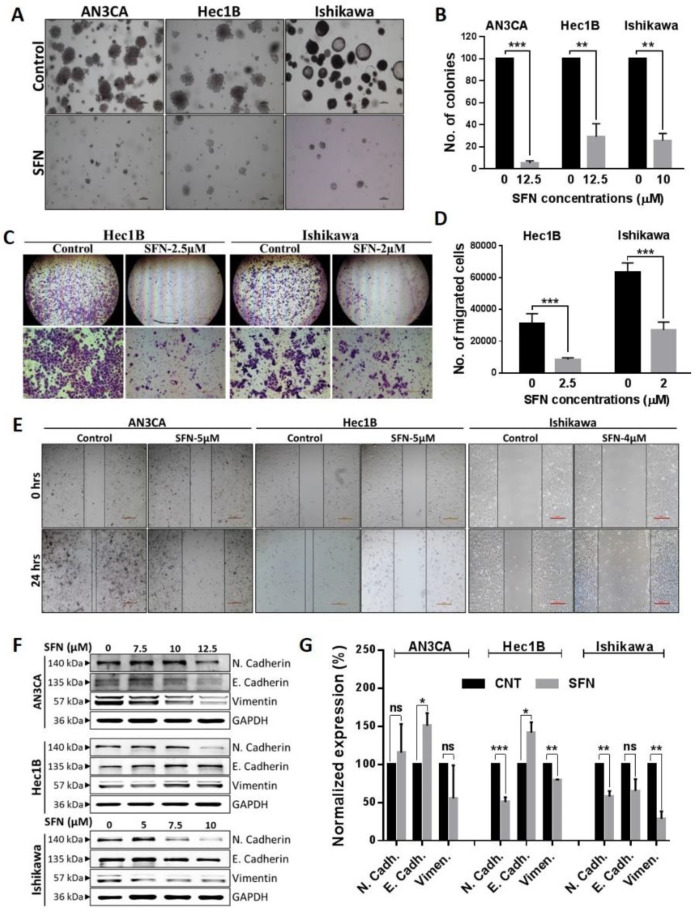
Sulforaphane inhibits endometrial cancer cell clonal growth, migration, and invasion. (**A**,**B**) A soft-agar colony formation assay was performed in endometrial cells treated with or without sulforaphane, and representative images were captured using an inverted microscope (**A**). Numbers of the colonies were counted using a Gelcount colony counter and the vehicle treated control was set to as 100%. Data are mean ± SD of three independent experiments and an unpaired *t*-test was used for statistical analysis (**B**). (**C**,**D**) Transwell invasion assays were performed in the endometrial cell with or without sulforaphane cells and microphotographs were taken at 4× (**C**). Cell counting of migrated or invaded cells are shown. Data are expressed as mean ± SD and Two-way ANOVA was used for statistical analysis. (**E**) A wound was introduced on a confluent monolayer of endometrial cancer cells and cell migration into the wound was monitored for 24 h in the presence and absence of sulforaphane. Images were taken at 10× magnification at 0 h and 24 h. Black solid lines denote the margins of the wound. (**F**,**G**) Endometrial cancer cells were treated with sulforaphane at the indicated concentrations for 24 h and analyzed for the expression of EMT markers by western blot. GAPDH was used as a reference gene. Representative blots are shown (**F**) and densitometric quantitation of protein expression levels are shown as fold changes with ± SE and student’s *t*-test was used for statistical analysis (**G**). “*p*” values are indicated * *p* ≤ 0.05, ** *p* ≤ 0.01, *** *p* ≤ 0.001when compared with respective control. SFN; sulforaphane. The whole western blot images of Figures please find in Appendix A.

**Figure 4 cancers-12-01273-f004:**
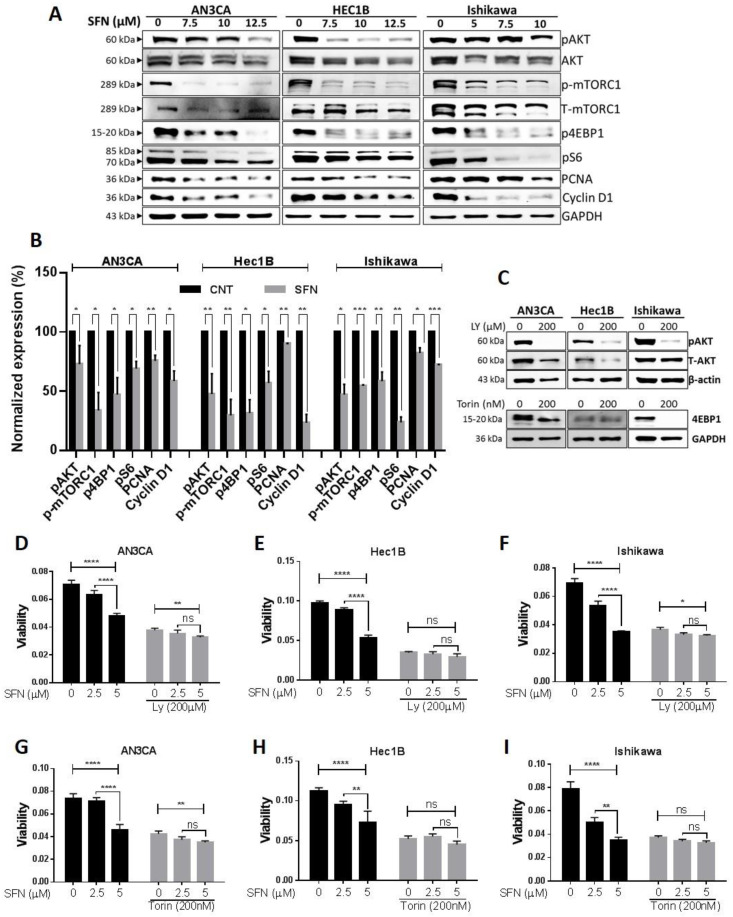
Sulforaphane inhibits endometrial cancer cell proliferation via inhibition of the PI3K-AKT-mTOR pathway. (**A**,**B**) Endometrial cancer cells were treated with the indicated concentrations of sulforaphane for 24 h, and protein isolated were analyzed for expression of PI3K-AKT-mTOR signaling markers by western blot. GAPDH was used as a loading control. Representative blots are shown (**A**) and densitometric quantitation of protein expression levels in control (vehicle treated) vs. sulforaphane treated endometrial cancer cells (12.5 µM of sulforaphane for AN3CA and Hec1B and 10 µM of sulforaphane for Ishikawa) are shown as fold changes with ± SE and *t*-test was used for statistical analysis (**B**,**C**) Endometrial cancer cells were treated with PI3K inhibitor (LY) and mTOR inhibitor (Torin) at indicated concentration for 4 h. Protein isolates were analyzed by western blot analysis to confirm the knockdown of PI3K downstream effector phosphorylation of AKT^Ser473^ and mTOR downstream effector phosphorylation of ^4EBP1Thr 37/46^. GAPDH was used as a control to correct for loading. Representative blots are shown. (**D**–**I**) Endometrial cancer cells were pretreated with LY (**D**–**F**) or Torin (**G**–**I**) or vehicle for 4 h prior to addition of sulforaphane for 24 h, and cell viability was analyzed by MTT. Cell viability (optical density) are shown as mean ± SD and Two-way ANOVA was used for statistical analysis. ‘*p*’ values are indicated * *p* ≤ 0.05, ** *p* ≤ 0.01, *** *p* ≤ 0.001, **** *p* ≤ 0.0001 when compared with respective control. SFN; sulforaphane. The whole western blot images of Figures please find in Appendix A.

**Figure 5 cancers-12-01273-f005:**
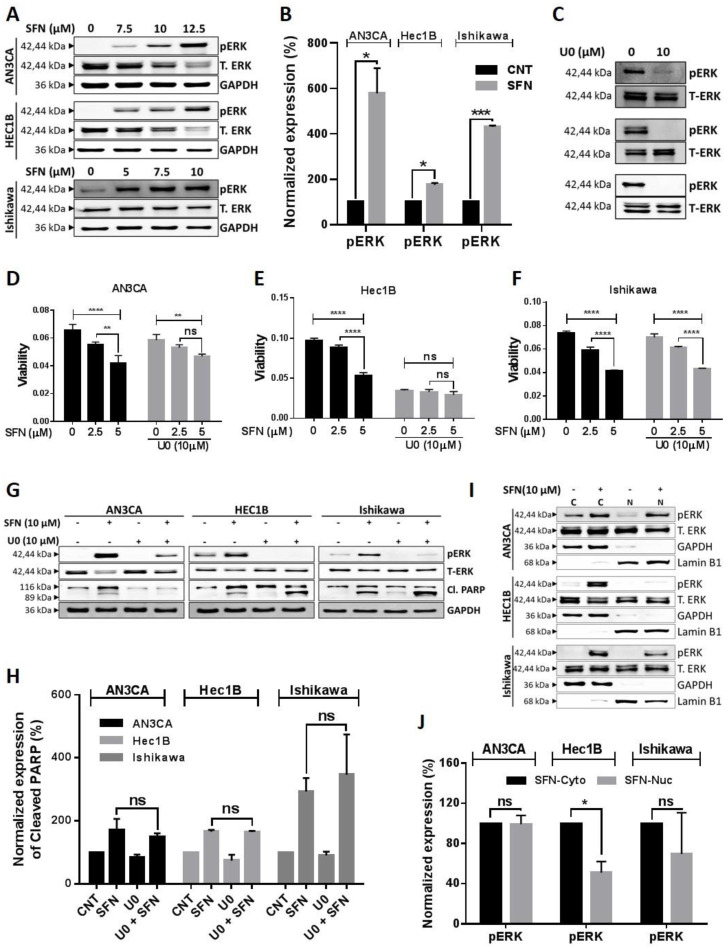
Sulforaphane induces phosphorylation of ERK/MEK and its cytoplasmic retention. (**A**,**B**) Endometrial cancer cells were treated with sulforaphane for 24 h and cell lysates were analyzed by western blot with antibodies against MEK-ERK signaling. GAPDH was used as a reference gene. Representative blots are shown (**A**) and densitometric quantitation of protein expression levels are shown as fold changes with ± SE and *t*-test was used for statistical analysis. (**C**) Endometrial cancer cells were treated with ERK inhibitor (U0) at indicated concentration for 4 h, and knockdown of pERK was confirmed by western blot analysis. Total ERK was used as a reference gene. Representative blots are shown. (**D**–**F)** Endometrial cancer cells were pretreated with U0 or vehicle for 4 h prior to addition of sulforaphane for 24 h, and cell viability was analyzed by MTT. Cell viability (optical density) are shown as mean ± SD and a Two-way ANOVA was used for statistical analysis. (**G**,**H**) Endometrial cancer cells were pretreated with indicated concentration of U0 or vehicle for 4 h before the treatment of sulforaphane for 24 h, and proteins were isolated. Expression of pERK, total ERK, and cleavage of PARP as an apoptosis marker were analyzed by western blot. GAPDH was used as a loading control. Representative blots are shown (**G**), densitometric quantitation of protein expression levels are shown as fold changes with ± SE, and *t*-test was used for statistical analysis of sulforaphane alone vs sulforaphane with pretreated U0 (**H**). (**I**–**J**) Cytoplasmic vs nuclear subcellular fractions were collected from endometrial cancer cells after the treatment of sulforaphane at the indicated concentrations. Expression of pERK was analyzed by western blot. GAPDH as a cytoplasmic marker and Lamin B1 as a nuclear marker was used as a loading control. Representative blots are shown (**I**), and densitometric quantitation of protein expression levels are shown as fold changes with ± SE and *t*-test was used for statistical analysis (**J**). “*p*” values are indicated * *p* ≤ 0.05, ** *p* ≤ 0.01, *** *p* ≤ 0.001, **** *p* ≤ 0.0001 when compared with respective control. SFN; sulforaphane. The whole western blot images of Figures please find in Appendix A.

**Figure 6 cancers-12-01273-f006:**
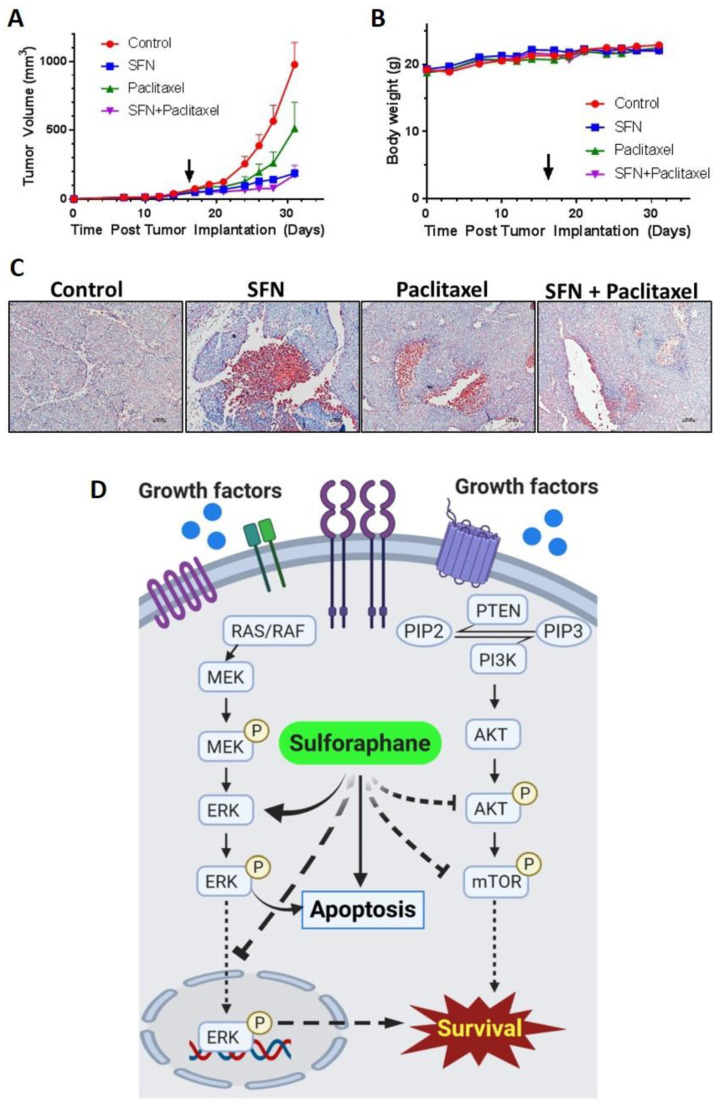
Sulforaphane, paclitaxel and the combination of sulforaphane and paclitaxel reduced tumor volume when compared to control. Severe combined immunodeficiency (SCID) mice bearing an average tumor volume of 50 mm^3^ Ishikawa xenografts were randomized into treatment groups which received daily 0.9% saline (control), daily 50 mg/kg sulforaphane, weekly 10 mg/kg paclitaxel, or daily 50 mg/kg sulforaphane in combination with weekly 10 mg/kg paclitaxel. Each drug was administered via intraperitoneal injection. The arrow denotes the inititation of treatment. *n* = 10 mice per group. (**A**) average tumor volume over time in groups treated with sulforaphane, paclitaxel, or sulforaphane and paclitaxel. (**B**) average body-weight of mice receiving each treatment regimen over the course of administration. *n* = 10 mice per group. (**C**) tumor tissues were analyzed by tunnel assay for apoptosis. Red color indicates positive signal. Representative images taken at 10× magnification are shown. (**D**) Schematic model for sulforaphane mediated endometrial cancer tumor inhibition. SFN; sulforaphane.

**Table 1 cancers-12-01273-t001:** Sulforaphane potencies (µM) and efficacies (maximal % growth inhibition) against human endometrial cancer cell lines.

Cell Line	Sulforaphane	IC_50_ +/− SE	Efficacy +/− SE
MFE280	24 h	13.50 +/− 3.44	35 +/− 2%
	48 h	21.17 +/− 1.59	44 +/− 1%
	72 h	36.99 +/− 7.01	53 +/− 1%
KLE	24 h	12.54 +/− 1.09	43 +/− 1%
	48 h	8.14 +/− 1.16	47 +/− 0%
	72 h	11.01 +/− 1.17	63 +/− 1%
Ishikawa	24 h	12.95 +/− 1.91	74 +/− 0%
	48 h	4.04 +/− 1.05	87 +/− 0%
	72 h	4.18 +/− 1.03	90 +/− 0%
Hec1B	24 h	9.23 +/− 1.06	39 +/− 4%
	48 h	9.13 +/− 1.09	68 +/− 4%
	72 h	9.14 +/− 1.08	78 +/− 3%
Hec1A	24 h	3.94 +/− 1.14	29 +/− 2%
	48 h	5.00 +/− 1.08	49 +/− 1%
	72 h	5.08 +/− 1.05	68 +/− 0%
MFE296	24 h	3.26 +/− 1.12	54 +/− 3%
	48 h	4.18 +/− 1.06	70 +/− 4%
	72 h	4.32 +/− 1.12	77 +/− 2%
AN3CA	24 h	0.90 +/− 1.21	14 +/− 3%
	48 h	0.84 +/− 1.06	39 +/− 1%
	72 h	0.75 +/− 1.06	57 +/− 2%

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
