# Peer review of "Preclinical Efficacy and Involvement of AKT, mTOR, and ERK Kinases in the Mechanism of Sulforaphane against Endometrial Cancer"

_cancers, 2020, doi:10.3390/cancers12051273_

Round 1
Reviewer 1 Report
Dear Authors,
please find all my comments, suggestions and doubts in pdf file.
Best regards,

Author Response
We are very thankful to reviewer for providing such in-depth critique of our manuscript. We tried our best to address all the point raised by him. These modifications helped us to improve our manuscript more scientific and publishable.
- Line 7: No 1 lacking.
Response: Added as suggested.
- Line 12: Running title might be better.
Response: Running title is modified as “Anti-cancerous Activity of Sulforaphane in Endometrial Cancer”.
- Line 13: Space lacking.
Response: Space is added between sulforaphane and exerts.
- Line 15: Provide a full name of cdc2 for the first time.
Response: We added full name of cdc2 as “cell division cycle protein 2 (cdc2)”.
- Line 17: Sulforaphane-induced.
Response: Added as suggested.
- Line 18: Vimentin expression.
Response: Added as suggested.
- Line 24-26: don't understand.
Response: Line 24-26 is modified to make it easier to understand and to reduce the total number of words in abstract (since limit is 200 word) as follows: “Sulforaphane-induced apoptosis-associated growth inhibition of Ishikawa xenograft tumors to a greater extent than paclitaxel, with no evidence of toxicity”.
- Line 27: treatment candidate.
Response: Added as suggested.
- Line 64: and.
Response: We corrected this spelling mistake.
- Line 84-93: Here, avoid presenting the results, just summarize methods used to evaluate it. Summarized results should be present in conclusions.
Response: As suggested, we modified line 84-93 as “we evaluated the anticancerous activity of sulforaphane and its mechanism of action. Proteomic analyses were performed to identify signaling mechanisms of the sulforaphane activities. Further, an endometrial cancer cell line xenograft study was performed to validate the in vitro findings”.
- Line 95: Inhibits.
Response: Modified as suggested.
- Line 100: growth of these cell lines was inhibited
Response: As suggested, we modified text as “demonstrated that growth of all the eight cell lines tested were inhibited by sulforaphane in a dose- and time-dependent manner.”
- Supplementary Fig. 1. Were the results significant, no statistical analysis presented at plots.
Response: We added statistical significance in a table form as Fig. S1b.
- Line 105: It will be nice to summarize the differences of cell lines chosen for the rest of experiments, e.g. Ishikawa cell line is hormone-sensitive, responding to estrogens which link with obesity is crucial, what about other cell lines?
Response: As suggested, we have added following text in the manuscript “All the three cell lines are type-1 endometrial cancer cell lines thus representing the most common form of endometrial cancer. There hormone receptor and differentiation statuses are representative of the wide range of endometrial cancers. The Ishikawa cell line is well differentiated, expresses the progesterone receptor (PR) and both forms of the estrogen receptor (ER), and is hormone-sensitive. HEC-1B is ER- and PR-negative and moderately differentiated, while AN3CA is ER- and PR-negative and undifferentiated”.
- 1 B & C: Why no statistics presented?
Response: As suggested, statistical significance was added to Fig. 1B & C in the main Figure.
- Fig 1F: explain why you used a different concentrations of SFN for Ishikawa cells.
Response: We added following text in the main manuscript on line 138: “Because Ishikawa was the most sensitive, we used slightly lower concentrations with a maximum 10µM sulforaphane to treat Ishikawa cells in comparison to the doses used for the other two cell lines.”
- Line 109: proliferation, not viability.
Response: Viability was replaced with proliferation.
- Line 114: use SFN instead a full name if possible.
Response: We respect reviewer’s comment, however, to avoid any kind of heterogeneity in the manuscript due to frequent use of sulforaphane we are leaving it as it was, this time.
- Line 121: “p” value should be always in italics.
Response: As suggested we italicized all ‘p’ in to ‘p’ throughout the manuscript.
- Line 127: BrdU is not cells viability, but proliferation.
Response: We modified the line as “sulforaphane-mediated inhibition of cell proliferation involves reduction………..”
- Line 138: There is a link to wikipedia?
Response: We removed hyperlink.
- Line 145: explain it.
Response: We modified text to add the following phrase at the end of the sentence; “ Further cyclin A expression was down-regulated in Hec1B and AN3CA cells, and upregulated in Ishikawa cells.
- Line 146: induces- when it is simple it is easier for the reader.
Response: As suggested we modified line as “Sulforaphane induces Endometrial Cancer Cell Death”.
- Line 151: is it a full name?
Response: Although JC-1 is very common name we added its full name as “JC-1 (tetraethylbenzimidazolylcarbocyanine iodide) assay” in the manuscript.
- Line 151: I do not agree, I suppose you mean mitochondrial membrane potential.
Response: We agreed with reviewer’s comment and replaced ‘metalloproteinase’ with ‘mitochondrial membrane potential’.
- 2G: Why the optical density calculations are not provided?
Response: As suggested we added densitometry analysis of Fig. 2G as ‘Fig. S2: Densitometry analysis of Sulforaphane induced apoptotic markers’.
- Line 212: t-test.
Response: As suggested, ‘t-test’ is modified as ‘student’s t-test’.
- Line 239: These
Response: Corrected for typo error.
- Line 239: provide full name of PCNA
Response: Added as ‘Proliferating cell nuclear antigen (PCNA)’.
- 4B: Is it SFN, what concentration?
Response: We apologies for our mistake. SHetA2 is replaced with SFN and concentration is added in Fig 4 legend as “sulforaphane treated endometrial cancer cells (12.5 µM of sulforaphane for AN3CA and Hec1B and 10 µM of sulforaphane for Ishikawa)”.
- 4 D-G: Why did you not compare the results with and without LY?
Response: Thank you for your kind suggestion. The main aim of this experiment was to confirm the involvement of PI3K-AKT pathway in sulforaphane induced cell death by determining if LY interfered with the dose-response effect observed for sulforaphane. Since we were not exploring any efficacy study in between sulforaphane and LY, we didn’t add statistical analysis of sulforaphane and LY.
- Line 276-277: Why you did not put all together in one figure?
Response: Thank you for your kind suggestion. The main study discussed in this figure (Figure 5 A-J) was about ERK1/2 or Phospho-ERK1/2, so it was not logical to put MEK1/2 or Phospho-MEK1/2 result together.
- Line 279: Provide a dose of U0.
Response: U0 dose was added in the manuscript as “dose (10 µM) of the ERK1/2 inhibitor, U0126 (U0)”.
- Fig 5C: In my opinion providing this figure is not necessary, similar figure is provided below, where cells are treated only with inhibitor (G)
Response: Figure 5C-F and 5G, are completely two different experiments.
Figure 5 C-F were performed to confirm the involvement of ERK1/2 pathway in sulforaphane mediated cell death. For the MTT experiment (Fig. 5D-F), endometrial cells were pretreated with ERK1/2 inhibitor for 4 hrs before sulforaphane treatment, so it was necessary to confirm the knockdown of ERK1/2 phosphorylation by inhibitors. In Fig 5C, proteins were isolated after 4 hrs of U0 treatment and that confirms the knockdown of ERK1/2 phosphorylation in the presence of inhibitors within 4 hrs.
Figure 5G represents the expression of proteins after pretreatment of U0 and then sulforaphane treatment for 24 hrs. Proteins were isolated after 24 hrs of treatment of sulforaphane.
- Fig 5 D-F: Similar question here, why you did not compare dose of SFN with and without inhibitor?
Response: Thank you for your kind suggestion. The main aim of this experiment was to confirm the involvement of ERK1/2 pathway in sulforaphane induced cell death by determining if U0 interfered with the dose-response effect observed for sulforaphane. Since we were not exploring any efficacy study in between sulforaphane and U0, we didn’t add statistical analysis of sulforaphane and U0.
- Fig 4G: Cleaved PARP was conducted before for 10uM SFN and different results were presented e.g. for AN3CA cell line in first figure (figure 2) the lower band was more present than the higher, here the results are contradictory? Could you explain it? Another antibody was used, another cell passage etc.? Moreover, usually cleaved PARP antibodies presents cleaved form at 89kDa- why did you choose to analyze both?
Response: Although in Fig. 4G the western blot result does not looks like to Fig. 2, our densitometry results is showing similar findings with significantly increased cleavage of PARP as shown in Fig. 2. Here we are showing the best representative blot for all the compound combinations.
Further we choose the Total PARP antibody (CST#9542) to analyze the effect of sulforaphane on both (Total and cleaved PARP) bands.
- Line 328 : explain why Ishikawa model
Response: We choose to utilize Ishikawa cell line for xenograft study because our review of the literature gave us the impression that this cell line is the most commonly used and accepted model.
- Line 333: data in table are generated at 1st may 2020- I really doubt how might it be possible for you, because I did obtain the review earlier?
Response: Table 4 is SAS system generated file. Automatically it takes the current date whenever we open the file.
- 6A: I do not understand, the control is statistically significant different from what?
Response: Control is statistically different with sulforaphane alone and combination of sulforaphane and paclitaxel groups. We removed the star from the control group to avoid any confussion. We have already provided the detailed analysis in Table S4.
- Fig 6D: Growth.
Response: Corrected for typo error.
- Line 353-354: These results are not presented here.
Response: We have removed the p value details.
- Line 392: Here, the results comparing viability of cells after treatment with and without inhibitors and SFN should be provided
Response: Thank you for your kind suggestion. As we mentioned earlier in our response (response no 31 and 35), the main aim of sulforaphane with or without inhibitor experiment was to explore the action mechanism not to explore any efficacy study in between sulforaphane and inhibitor, so there will be confusion if we add anything here.
- Line 395: d?
Response: Deleted.
- Line 427: What about inhibitors?
Response: Inhibitor details are added in the manuscript as “PI3K inhibitor, LY294002 (#9901); mTOR inhibitor, Torin2 (#14385); and ERK1/2 inhibitor, U0126 (#9903) were purchased from Cell Signaling Technology (Danvers, MA, USA).
- Line 430: Typo error.
Response: Corrected for typo error.
- Line 430: this is really small number of cells, are you able to provide the confluency of cells when induced?
Response: We apologies for typo error and corrected cell density “4000-6000” was added in the manuscript.
- Line 438: What type of control was used in all experiments, vehicle or non-treated cells? What medium was used for treatment with or without FBS or other supplements?
Response: To clarify we added following lines in the Cell lines, culture conditions, and chemicals section of Materials and methods as “In all of the experiments, cells treated with vehicle only were considered as the control and all treatments were given in medium supplemented with 10% FBS”.
- Line 465: Typo error.
Response: Corrected.
- Line 522: equal of approximately? it is impossible to use both of them, provide an equal amount of protein used per well. If it is different for antibodies, provide it.
Response: The word “Approximately” is deleted.
- Line 527: Thrice?
Response: Line is modified as “The membranes were incubated with primary antibodies at 4˚C, overnight followed by three times washing with TBST”.
- Line 543: here is a contradictory information, above you used 1:6000 or 1:5000
Response: We corrected for secondary antibody dilution as “Secondary antibodies were used at a dilution of 1:5000-6000.
- Line 553: define: one or twice?
Response: One is deleted.
- Line 560: 5X104 cells?
Response: It is corrected as 2.5X104 cells.
- Line 579: provide the number of ethical compliance.
Response: We have added the ethical compliance as “(Protocol #18-010-ICH-H)”.
- Line 579: provide information how mice were sustained
Response: The information how mice were sustained were added as “Female 5 week old NOD/SCID mice were obtained from Taconic Biosciences, Inc and were maintained under a 12-hour light/dark cycle under pathogen-free conditions, using sterile food and water within the cage. After acclimation period of 72 hrs………….
- Line 595-596: I cannot find data obtained from this tissues?
Response: This Sentence is deleted from the manuscript.
- Line 631: Spec?
Response: ‘Spec’ is replaced with ‘spectrometry’.

Reviewer 2 Report
This manuscript addresses the anti-cancer activity of a naturally-occurring compound, sulforaphane, which is found in cruciferous vegetables, in endometrial cancer. Sulforaphane has been associated with chemopreventative and anti-cancer activity in other cancer types through a variety of mechanisms (cell cycle inhibition, apoptosis, and epigenetic modulation). Importantly, this compound had been deemed safe in clinical studies. The authors present a carefully executed study to interrogate the impact and mechanism of action of sulforaphane on a panel of endometrial cancer cell lines. Overall, the data support the conclusions. I have a few comments that would strengthen the study and aid with data interpretation.
Comments:
- Sulforaphane clearly has a time- and dose-dependent impact on multiple types of endometrial cancer cells. However, the concentrations used are quite high – in 5-10 μM range. Are these doses achievable in humans?
- Figure 1, Line 139-140: the authors state that, “Cdc2 is phosphorylation in the G2 phase and its de-phosphorylation in the M phase leads to the prevention of entry into mitosis [32].” However, it is widely accepted that de-phosphosphorylation of Cdc2 by the phosphatase CDC25C allows cells to enter into M phase, rather than arrest in G2 as the authors state. In the discussion (lines 361-362), the authors correctly explain the signaling events that lead to G2 cell cycle checkpoint maintenance.
- Figure 1: The authors observed increased phosphorylation of Histone H3 at Ser 10, a canonical marker of mitosis. Please explain why there is both an increase in Cdc2 phosphorylation, indicative of maintenance of the G2/M checkpoint, and Histone H3 phosphorylation, indicative of progression into M phase. Was phosphorylation of CDC25C assessed? Or PI staining for DNA as an indicator of M-phase progression, as was reported in Meng et al., Gynecol Oncol 2013: 128(3):461?
- Figure 4B: The treatments are labeled CNT and SHetA2 for the quantitation of data in Fig. 4A. Is this a typo or are the wrong data displayed? Also, which concentration of drug was chosen for the quantitative analysis in Fig. 4B?
- Figure 4A & C: Why were the timepoints different between these two experiments? This experiment should be repeated with sulforaphane and the PI3K inhibitor on the same blot for the same duration of treatment. In addition, the compound used to inhibit PI3K is known to have off-target effects on other kinases, such as DNA-PK, and it was used at an exceptionally high concentration (200 μM). Thus, it is difficult to definitively conclude from these experiments that sulforaphane acts through the PI3K/Akt/mTOR signaling pathway. A more selective PI3K inhibitor should be used for these studies.
- Cell viability studies in Figures 4&5 with combination treatments: The legends state that the data are expressed as % viability, but the y-axis does not go from 0-100% (as in Figure 1B, C). This makes the data very difficult to interpret.
- Figure 5 I & J: include total ERK1/2 levels in the cytoplasmic & nuclear fractions.
- There are several spelling and syntax errors throughout the manuscript.
Author Response
Thank you very much for providing a comprehensive appraisal of our manuscript. Your critics helps us to revise our work and improve it scientifically and significantly. We are providing herewith an explanatory rebuttal to your comments point by point, which are as followings:
Comments: Reviewer 2
- Sulforaphane clearly has a time- and dose-dependent impact on multiple types of endometrial cancer cells. However, the concentrations used are quite high – in 5-10 μM range. Are these doses achievable in humans?
Response: We acknowledge that the concentrations of sulforaphane added to our cell culture experiments is above the physiologically achievable methods. Our cell culture model has the limitation of being only a 24 to 72 hours exposure, whereas the exposure time in clinical trials an in vivo experiments is weeks to months. Thus, higher levels are needed to induce a response in a short term cell culture experiment compared to in vivo studies. Therefore, our animal model provides a critical validation of the cell culture experiments in this study.
We already addressed this concern on line 405 of the discussion: Significant inhibition of Ishikawa endometrial cancer cell line xenograft growth by sulforaphane in association with induction of apoptosis provided in vivo validation of our cell culture results.
- Figure 1, Line 139-140: the authors state that, “Cdc2 is phosphorylation in the G2 phase and its de-phosphorylation in the M phase leads to the prevention of entry into mitosis [32].” However, it is widely accepted that de-phosphosphorylation of Cdc2 by the phosphatase CDC25C allows cells to enter into M phase, rather than arrest in G2 as the authors state. In the discussion (lines 361-362), the authors correctly explain the signaling events that lead to G2 cell cycle checkpoint maintenance.
Response: We modified line 139-140 “Cdc2 is phosphorylation ….” as “Cdc2 is phosphorylated in the G2 phase leading to cell cycle arrest and prevents entry into mitosis but is de-phosphorylation in the M phase leading progression to mitosis [32].
- Figure 1: The authors observed increased phosphorylation of Histone H3 at Ser 10, a canonical marker of mitosis. Please explain why there is both an increase in Cdc2 phosphorylation, indicative of maintenance of the G2/M checkpoint, and Histone H3 phosphorylation, indicative of progression into M phase. Was phosphorylation of CDC25C assessed? Or PI staining for DNA as an indicator of M-phase progression, as was reported in Meng et al., Gynecol Oncol 2013: 128(3):461?
Response: With this criticism, we realized our mistake of calling the sulforaphane-induced cell cycle arrest G2 instead of G2/M phase. Our PI staining Flow Cytometric analysis does not distinguish G2 from M phase arrest. Further, Phospho-histone 3 (pH3) is also a marker of mitosis and cell cycle arrest during the G2/M phase.
Thus, we have changed the term ‘G2 phase’ arrest to ‘G2/M’ phase arrest throughout the manuscript. We haven’t assessed CDC25C phosphorylation since we noticed an increase in Cdc2 phosphorylation which is indicative of its inhibition, and of maintenance of the G2/M checkpoint.
- Figure 4B: The treatments are labeled CNT and SHetA2 for the quantitation of data in Fig. 4A. Is this a typo or are the wrong data displayed? Also, which concentration of drug was chosen for the quantitative analysis in Fig. 4B?
Response: In Fig 4B, treatment label “SHetA2” is a typo error. We replaced SHetA2 by SFN. Also, for the quantitative analysis, the selected concentration of drug was added in the fig. legend as “densitometric quantitation of protein expression levels in in control (vehicle treated) vs. sulforaphane treated endometrial cancer cells (12.5 µM of sulforaphane for AN3CA and Hec1B and 10 µM of sulforaphane for Ishikawa) are shown.
- Figure 4A & C: Why were the timepoints different between these two experiments? This experiment should be repeated with sulforaphane and the PI3K inhibitor on the same blot for the same duration of treatment. In addition, the compound used to inhibit PI3K is known to have off-target effects on other kinases, such as DNA-PK, and it was used at an exceptionally high concentration (200 μM). Thus, it is difficult to definitively conclude from these experiments that sulforaphane acts through the PI3K/Akt/mTOR signaling pathway. A more selective PI3K inhibitor should be used for these studies.
Response: The graphs in Figure 4A and C, are two completely different experiments. Figure 4A represents the expression of proteins after sulforaphane treatment for 24 hrs.
Figure 4 C-I were planned to confirm the involvement of PI3K-AKT-mTOR pathway in sulforaphane-mediated cell death. Since endometrial cells were pretreated with PI3K and mTOR inhibitor for 4 hrs before sulforaphane treatment in the MTT experiment (Fig. 4D-I), it was necessary to confirm that the inhibitors effectively knockdown the expression of these proteins before sulforaphane treatment was initiated. Figure 4C confirms knockdown of PI3K downstream effector (AKTSer473phosphorylation) and mTOR downstream effector (4EBP1Thr 37/46 phosphorylation) in after 4 hours to treatment with the inhibitors.
We agreed with the comment regarding the target effects of inhibitors, that is why we chose two different inhibitors (PI3K-LY294002 and mTOR inhibitor-Torin2) to confirm the role of PI3K-AKT-mTOR pathway in sulforaphane mediated cell death and we observed similar results in both cases.
- Cell viability studies in Figures 4&5 with combination treatments: The legends state that the data are expressed as % viability, but the y-axis does not go from 0-100% (as in Figure 1B, C). This makes the data very difficult to interpret.
Response: We modified “% cell viability” into “cell viability (optical density)” in the figure 4 & 5 legends.
- Figure 5 I & J: include total ERK1/2 levels in the cytoplasmic & nuclear fractions.
Response: As suggested, we have added western blot result of cytoplasmic & nuclear fractions of total ERK1/2 levels in the Figure 5 I. I have also added the full blots in to the supplementary file.
- There are several spelling and syntax errors throughout the manuscript.
Response: We revised whole manuscript for spelling and syntax corrections.

Round 2
Reviewer 2 Report
The authors have addressed all of my critiques in the revised manuscript. One additional minor revision should be made prior to publication:
In the legends of Figures 4G-I and 5 D-I, please make it clear that cells were pretreated for 4 hrs with the inhibitors prior to addition of SFN. The authors mention this in the response to reviewers, but the legend states that the cells were treated with the combination for 24h.
Author Response
We are thankful to reviewer for pointing these mistakes. As suggested we modified legend of Fig. 4 (D-I) and Fig. 4 (D-H) as given bellow:
Figure 4 (D-I): Endometrial cancer cells were pretreated with Ly (D-F), or Torin (G-I) or vehicle for 4 h prior to addition of sulforaphane for 24 h, and cell viability was analyzed by MTT.
Figure 5 (D-F): Endometrial cancer cells were pretreated with U0 or vehicle for 4 h prior to addition of sulforaphane for 24 h, and cell viability was analyzed by MTT.
Figure 5 (G-H): Endometrial cancer cells were pretreated with indicated concentration of U0 or vehicle for 4 h before the treatment of sulforaphane for 24 h, and proteins were isolated.
